# Calcium Sulfate and Plasma Rich in Growth Factors Enhance Bone Regeneration after Extraction of the Mandibular Third Molar: A Proof of Concept Study

**DOI:** 10.3390/ma14051126

**Published:** 2021-02-27

**Authors:** María Huchim-Chablé, Roberto Sosa-Martínez de Arredondo, José Alberto Rivero-Navarrete, Celia Mendiburu-Zavala, Rubén Cárdenas-Erosa, Ricardo Peñaloza-Cuevas

**Affiliations:** 1General Dentistry Private Practice; Av. Itzaes # 252 X 29 Colonia García Ginerés, Mérida 97070, Yucatán, Mexico; azucenita26@hotmail.com; 2Private Practice, Central Odontológica of Yucatán; 26 Street # 218-B X 27 and 29, Colonia García Ginerés, Mérida 97070, Yucatán, Mexico; codyscp@yahoo.com.mx; 3Statistics Research Department, Universidad Autónoma of Yucatán, 55 Street X Circuito Colonias y Fraccionamiento del Parque, Mérida 97159, Yucatán, Mexico; rnavar@correo.uady.mx; 4Faculty of Dentistry, Universidad Autónoma of Yucatán (UADY), 61st Street # 492 x Av. Itzáes, Colonia Centro, Mérida 97000, Yucatán, Mexico; mzavala@correo.uady.mx (C.M.-Z.); cerosa@correo.uady.mx (R.C.-E.)

**Keywords:** bone regeneration, graft, calcium sulfate, PRGF

## Abstract

The aim of this study was to evaluate the mixture of Calcium Sulfate and Plasma Rich in Growth Factors (CaSO_4_ + PRGF) as a bone-graft substitute in extracted mandibular third molar (MTM) alveoli during a 4-month period. Bilateral MTM extractions were performed in 10 patients (18–25 years) at the Oral-Surgery-Clinic of the Universidad Autónoma de Yucatán (UADY). A CaSO_4_ + PRGF mixture was placed in the right alveolus (Experimental Group (EG)) and a natural blood clot in the left (Control Group (CG)). Monthly X-ray controls were performed using a gray scale to measure Bone Regeneration (BR). A non-parametric Sign Test was used to evaluate Radiopacity/Bone Regeneration (Ro/BR) over 4 months, and a Friedman’s non-parametric test was used for intra-group analysis over these months. The study was approved by the Centro de Investigaciones Regionales (Dr. Hideyo Noguchi, UADY Bioethics Committee, ID 0026-2015). Using a non-parametric test of the sign, the EG showed significant difference of Ro/BR between groups *p* = 0.002 (*p* < 0.05). Significant differences were observed in all quadrants and areas *p* = 0.002 (*p* < 0.05) except in area A in month 4 (*p* = 0.016), which could be explained by its being the closest to native bone. EG CaSO_4_ + PRGF showed a higher degree of bone regeneration compared to CG.

## 1. Introduction

In the maxilla 25% bone resorption after tooth extraction is characterized by height and width loss in the first year and that reaches 40% in the third. Resorption begins mainly on the buccal surface, as it is thinner and more fragile [1,2,3]. 

In recent years, there has been increased interest in finding better biomaterials to work as bone substitutes to prevent alveolar surface collapse, both vertically and horizontally, and prevent soft-tissue loss to ensure optimal aesthetic results for future rehabilitations [1,4,5,6]. Solis et al. (2009) mention that an ideal bone substitute should be able to repair anatomy and function and speed up ossification. Furthermore, it should be easy to handle, biocompatible, sterilizable, inexpensive and widely available [1,6]. It should also promote osteogenesis (the ability to generate bone tissue) osteoconduction (the ability to function as a matrix) and osteoinduction (the ability to cause cell differentiation) [1,7,8].

Grafts come in various classifications. An autograft (from the same individual) is unique because it complies with the 3 regeneration mechanisms in a natural way. Intraorally, it is obtained from the chin, retromolar region, nasal spine, exostoses, ascending mandibular ramus or lower edge of the jaw. Its advantages are that it avoids skin scars and is close to the recipient area. Extraorally, it can be taken from the iliac crest, femur, tibia, fibula, humerus or, ribs. However, this type of graft increases morbidity and requires longer surgical time [1,6,7,8,9,10,11,12,13,14]. An allograft is taken from a bone bank (corpse tissue) and has osteoinductive and osteoconductive properties depending on how it is processed: frozen, freeze-dried (cold drying), freeze-dried-demineralized or bone irradiated [1,6,7,8,9,10,11,12,15,16]. A xenograft is obtained from other species (bovine, porcine or equine). It has osteoconductive properties (Bio-Oss^®^ has an intimate contact of 36.7% with the bone of the surrounding surface) [1,6,7,8,9,10,11,12,17]. Alloplastics are synthetic (inert) and can be made from ceramic, hydroxyapatite polymers, tricalcium phosphate and CaSO_4._ These materials are used alone or in combination with other materials. They provide osteoconductive properties since hydroxyapatite is the fundamental component of bone [1,6,7,8,9,10,11,12,18,19,20]. The grafts can be placed in the receiving area in the form of a block or particulate [6,7,8,9,10,11,12,21,22,23].

Calcium sulfate (CaSO_4_ or “Plaster of Paris”) is an alloplastic material that has been used for over 100 years and the first to be grafted as a filler for bone defects in medicine and dentistry. It is currently used as a barrier for Guided Tissue Regeneration (GTR) [24,25,26]. It is biocompatible, does not induce inflammatory reactions, can repair and heal the bone, it is easy to obtain, osteoconductive, economical, abundant in nature and sterilizable. It is absorbed by dissolution in 8 weeks, depending on the volume and site of implantation. It is a crystalline salt that, when heated to 110 °C, loses water in a process known as calcination. Lebourg and Biou (1961) placed CaSO_4_ in the alveoli of third molars and observed complete reabsorption of the graft in the 3–4 weeks after extraction. In addition, they noted that bone repair in the experimental area was faster than in the control one [24,27,28,29]. 

Plasma Rich in Growth Factors (PRGF) is an endogenous therapeutic technology used in regenerative medicine thanks to its potential for stimulating and accelerating tissue healing through the growth factors found in platelets, which are universal initiators of almost every regeneration process. In dentistry, the first studies with PRGF were in the treatment of post-extraction defects because the regeneration of alveolar bone and surrounding soft tissue was necessary to ensure the future success of dental implants [30,31]. PRGF, which is 100% autologous and biocompatible, is obtained from the patient’s blood, thus eliminating the possibility of disease transmission and having a convenient obtainment cost [32,33,34]. The advantages of using PRGF include lower post-operative inflammation, faster soft and hard tissue healing (obtaining a better bone quality) and minimizing the risk of infection [33,35]. PRGF promotes angiogenesis, stimulates cell migration, increases proliferation and stimulates paracrine and autocrine secretion [36]. 

Platelets contain several growth factors. Platelet derived growth factor (PDGF) indirectly promotes angiogenesis through macrophages by chemiotaxis and facilitates the formation of type I collagen. Beta-transformation growth factor (TGF-BETA) participates in the proliferation and differentiation of mesenchymal cells, synthesis of collagen by osteoblasts, inhibition of the formation of osteoclasts and proliferation of epithelial cells in the presence of other factors. Fibroblast growth factor (FGF), which favors the proliferation and differentiation of osteoblasts, inhibits osteoclasts and can be found in saliva. Insulin-like growth factor (IGF), which is synthesized in the liver, intervenes in apoptosis and so can induce survival and send protective signals to different non-neoplastic cell types. Vascular endothelial growth factor (VEGF) is the most powerful as it controls the behavior of endothelial cells and the hyperpermeability of blood vessels; Epidermal growth factor (EGF) is mitogenic, proapoptotic and chemotherapeutic, and it participates in the differentiation of epithelial, renal, glial and fibroblast cells [30,32,33,37].

When CaSO_4_ + PRGF is mixed, an exothermic reaction occurs that promotes the activation of plasma growth factors, inducing angiogenesis, in the alveolus or bone defect, due to the presence of bone in the lower and lateral portion. The graft also acts as a barrier to prevent the migration of epithelial cells from the mucous membrane to the grafted area, thereby maintaining the necessary space for cell repopulation or bone formation [22]. The aim of this study was to evaluate the mixture of Calcium Sulfate and Plasma Rich in Growth Factors (CaSO_4_ + PRGF) as a bone-graft substitute in extracted mandibular third molar (MTM) alveoli, over a 4-month period.

## 2. Materials and Methods

An analytical, prospective, experimental and longitudinal study was performed. The sample consisted of potential patients aged 18 to 25 years, male or female, who attended the Oral Surgery Clinic in the Autonomous University of Yucatan and met the inclusion criteria: two MTMs in vertical position, according to Winter’s classification [38] without active infection. All 46 patients were clinically examined according to NOM-013-SSA2-2006 (Norma Oficial Mexicana) for prevention and control of oral diseases [39] and assisted by X-ray orthopantomography (Orthopantomograph OP200 D^®^ Orthoceph^®^, Nahkelantie, Tuusula, Finland). Ten patients met the inclusion criteria for the sample: 9 women and 1 man. All procedures were performed by the same operator (PRGF preparation, surgical procedure and CaSO_4_ spheres). The MTMs were extracted, and CaSO_4_ + PRGF (EG) was immediately placed in the right alveolus (for convenience), whereas the left was managed in a “traditional” manner (healing and physiological blood clot formation (CG)).

For grafting, CaSO_4_ (MDC^®^ Dental, Zapopan, Jalisco, México) spheres with a diameter of 2-2.5 mm were made, sterilized at 123 °C via dry heat sterilization (LORMA M-08^®^ Mexico City, Mexico) for 40 min. The patient’s blood was then obtained using the BTI^©^ Plasma Transfer Device kit (PTD^®^, BTI Biotechnology Institute, Vitoria-Gasteiz, Álava, Spain) and the Endoret^®^ BTI (BTI Biotechnology Institute, Vitoria-Gasteiz, Álava, Spain) centrifuge at 2000 rpm for 8 min. From 9 cc of whole blood, 2 cc of PRGF was obtained in a sterile stick, adding 50 µL/cc of calcium chloride (PRGF activator) and CaSO_4_ spheres until a gummy consistency was obtained (CaSO_4_ + PRGF).

### Lower Third Molar Extraction and Graft Placement

Under an asepsis and antisepsis protocol, truncular anesthesia with Dentocain (mepivacaine & epinephrine 2%, Zeyco^®^, Zapopan, Jalisco, México) was applied to the lower dental nerve, 5.4 mL (3.6 mL at truncular level and 1.8 mL on the vestibular face). A full-thickness triangular flap was created. Using a low-speed handpiece (Medidenta^®^ Las Vegas, NV, USA) with sterilized water irrigation and a 703 milling bur (Hager & Mesinger^®^ GMBH, Deu NEUSS, Germany) osteotomy and odontosection, elevation (No. 3 elevator) (Dental USA^®^, Chicago, IL, USA) extraction of the third molars (17 and 32) cavity cleaning with a Lucas curette (Dental USA^®^, Chicago, IL, USA) and hemostasis were performed. The CaSO_4_ + PRGF graft was placed in the right alveolus leaving the left side with the natural clot. Both were sutured with 4-0 silk (Ethicon^®^ US LLC, Cincinnati, OH, USA) (Figure 1).

Immediate post-surgical periapical X-rays were carried out (Satelec X-Mind DC^®^ X-ray equipment, Burnlea Grove, Birmingham, UK). Post-operative indications and medication were given: one 20 mg capsule of Meloxicam every 12 h for 5 days, one 10 mg Ketorolac tablet every 8 h for 5, one 300 mg clindamycin capsule 8 h for 5 days and Bexident Gums gel (ISDIN^®^, Barcelona, Cataluña, Spain) after toothbrushing (3 times a day) for one month. Sutures were removed 7 days after the procedure and photographic and periapical X-ray records were taken to assess the degree of closure and healing of the surgical site.

To measure bone regeneration radiographically (Ro/BR), a homemade grey standardized aluminum scale (GSAS) was used. Each step of the scale was numbered from 1 to 6 corresponding to a difference of 16.66%, until a 100% recording the results in Table 1 (Figure 2).

For the systematic evaluation of the periapical X-rays, a diagram of the alveoli was designed and divided into four quadrants: I to IV. I and IV represented the distal quadrants and II and III the mesial. Each quadrant was divided into three areas (A, B and C) from the periphery towards the center of the alveolus (Figure 3).

For the analysis, a non-parametric Sign Test (study sites in the same patient with related samples) was used to compare the degree of Radiopacity/Bone Regeneration (Ro/BR) from the Experimental Group (EG) against the Control Group (CG) over 4 months. Friedman’s non-parametric test was also used for intra-group analysis that was carried out over months with the help of the SPSS package version 20.0 (IBM^®^ New York, NY, USA). Results were considered with a level of significance *p* < 0.05. The study was approved by the Bioethics Research Committee, Centro de Investigaciones Regionales (Dr. Hideyo Noguchi, ID 0026-2015). Informed consent was obtained from all individual participants in the study.

## 3. Results

A total of 20 MTMs were extracted, 10 from each side. Each alveolus underwent a periapical X-ray at the end of the surgical procedure and then at the first, second, third and fourth post-extraction months to observe Ro/BR in both groups.

Monthly radiographic evaluation comparing the groups are illustrated in Figure 4, Figure 5 and Figure 6. The Experimental Group showed a higher Ro/BR. 

Monthly significant difference, *p* = 0.002 was obtained with a sign test for Ro/BR between groups. In the Friedman test, a value of *p* = 0.0001 showed significant intra-group differences in both EG and CG over time. Ro/BR was observed to have increased over time in both groups (Table 2).

When comparing group quadrants, the sign test, showed a significant difference for each test as *p* values oscillated between 0.004 and 0.002 (Table 3).

For the areas of the groups, the sign test showed a significant difference for each test as *p* values oscillated between 0.004 to 0.002, except in area A at month 4 (*p* = 0.016) (Table 4).

## 4. Discussion

After dental extractions, bone resorption and remodeling of between 4.0 and 4.5 mm was reported, a situation that poses challenges for a subsequent oral rehabilitation. Bone regeneration has become an important topic in the reconstruction of alveolar architecture for implant support. Different methods were performed, such as autografts (gold standard) allografts, xenografts, and alloplastics [25,40,41,42].

In clinical models for BR, a blood clot is used as a control, demonstrating that when biological elements or factors converge negatively (poor bone quality or large volume defect) the presence of the blood clot alone, without any type of graft, is insufficient to achieve bone neoformation; similarly, the ability to repair bone may be limited by deficiencies in blood supply as we observed in the study [10,43,44].

Different biomaterials are used for bone regeneration. In line with this study, Jeong et al. exalted the properties of bioactive calcium phosphate materials as well as López et al. demonstrated the use of CaSO_4_ as a regenerative material in post-extraction alveoli by showing a minimal inflammatory response and rapid absorption, unlike other regeneration materials [25,45].

Previously in the group, Huchim et al. (2017) evaluated CaSO_4_ spheres alone as an osteoconductive material and as an accelerator of bone regeneration in a chronic periapical lesion in a 49-year-old female patient with post-surgical periapical radiographic control at 1, 2, 8, and 17 weeks. The results demonstrated the reabsorption of CaSO_4_ at week 8 as reported in the literature, and at week 17 we observed a radiopacity that was 90–95% like that of the surrounding bone, which led us to conclude that CaSO_4_ is a biocompatible material suitable for bone regeneration. This study coincides with the current one since at the fourth month the radiopacity oscillated between 83.4 and 100% [24,27,28,29,46].

Eda et al. (2017) compare PRGF (Anitua’s protocol, BTI Institute of Biotechnology, Vitoria, Spain) and Platelet Rich Plasma (PRP) using Marx’s protocol double-centrifuge method for bone regeneration in rat calvaria by carrying out a histological examination and a microteleconference. PRGF promoted increased bone volume compared with PRP. Histologically, the observed bone formed at 4–8 weeks in the PRGF group. In addition, the PRP group showed numerous inflammatory cells in comparison with the PRGF group. Since between 4 and 8 weeks the bone regeneration becomes evident, it coincided with the results found by us [47].

In three male guinea-pig groups (10 each), Vellejos et al. (2018) used G1 Calcium sulfate, G2 Xenograft (bovine) and G3 negative control (no bone substitute). After lower-right central-incisor extraction, the alveoli of G1 and G2 were filled with graft. Standardized X-rays after exodontia and 40 days were taken. Measurements were compared at the mesial, cervical, and distal level alveolus and showed significant differences: mesial *p* = 0.025 and cervical *p* = 0.043 and bone regeneration *p* = 0.019. They observed that that grafts worked as osteoconductors, also producing an anti-inflammatory response. The results of this study are like ours because *p* = 0.002 was obtained for BR, and according to quadrants II and III, which represent the alveolus mesial surface, *p* = 0.002 was obtained during 1–4 months. However, our percentage of BR was higher in quadrants III and IV and in area A (apical zone of the alveolus), which means that BR went from the periphery to the center of the alveolus [48].

Moreno et al. (2010) extracted retained MTMs in patients between the ages of 16 and 17, placing on right (experimental) side absorbable hydroxyapatite (HAR) and CaSO_4_ and on left (control) only a blood clot. X-ray evaluation at 4 months showed that concentration of bone trabeculae on the experimental side was greater than on the control side. Bone alveolar density *p* = 0.00001, like our study, showed a statistically significant high difference [49].

Shi et al. (2007) applied CaSO_4_/PRP and blood clot to the anterior alveolus jaw of a canine model, and CaSO_4_/PRP and CaSO_4_ alone to the posterior jaw. Results showed that adding PRP improved wound healing. Computed tomography scans were performed 1 day and 8 weeks after extraction to detect changes in ridge height. Statistically CaSO_4_/PRP significantly reached higher BR values vs. those of the control group at 2 week––only *p* = 0.036. However, in our study we obtained a *p* = 0.002 until the first month [50].

Serafini et al. (2020) reported a case of post-extraction alveolar ridge preservation with the placement of three previously bent leukocyte- and platelet-rich fibrin (L-PRF) membranes and clinical controls at 14, 28 and 60 days. Three months later, histomorphometric analysis of the grafted site was performed and newly formed trabecular bone was observed. They suggest that the use of L-PRF could achieve good results in dimension, bone quality and soft tissue healing [51].

Anitua et al. (2015) compared a PRGF clot and PRGF-fibrin membrane in the alveolus to a blood clot alone in 60 patients after the first mandibular molar was extracted. Clinical results showed a significant difference in pain, inflammation and tissue healing between the groups on days 3–7. Tomographic result after 10-12 weeks showed significantly higher bone density in the PRGF group (96.7%) vs the control (45.5%). These results agreed with ours in the third month of radiographic control in the EG (83.4–100%) and CG (49.99–66.64%) for bone regeneration [52,53].

Zhou et al. (2018) compared Choukroun platelet-rich fibrin (PRF), autologous bone, Choukroun PRF combined with autologous bone and a control group in 36 healthy New Zealand rabbits (jawbone defects created). Three rabbits per group were sacrificed at 2, 8 and 12 weeks after surgery. Radiographic results showed more bone regeneration in Choukroun PRF, autologous bone and Choukroun PRF combined with autologous bone vs. the control group. They highlight the economic savings of obtaining autologous bone; however, despite the fact that it is cheaper than xenografts or allografts and easy to obtain at the time of the patient’s surgery, it increases the possibility of infections and post-operative pain compared to the use of a low-cost and easy-to-obtain alloplastic material like CaSO_4_. It also avoids the need for a second surgery. However, the radiographic results showed radiopacity compatible with bone regeneration after the first month [54,55].

There are many surgical techniques aimed at preventing changes in the post-extraction alveoli. However, treatments requiring autografts, allografts and xenografts increase morbidity and cost, while CaSO_4_ is affordable and accessible. For this reason, the application of CaSO_4_ + PRGF grafting in the alveoli of MTMs showed advantages in the speed of bone formation, osteoconductive capacity, rapid healing, and less inflammation [22,23,56].

## 5. Conclusions

The combination and placement of the CaSO_4_ + PRGF graft in alveoli showed higher Ro/BR velocity and soft tissue scarring as well as less post-operative pain and faster recovery. It is therefore considered to be a suitable material for BR, which also proved to be easy to acquire and inexpensive.

## Figures and Tables

**Figure 1 materials-14-01126-f001:**
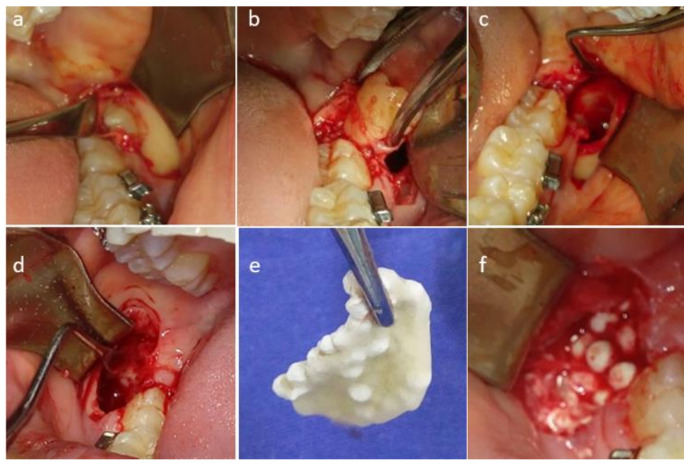
Surgical procedure: Control Group: (**a**) Full-Thickness triangular flap; (**b**) Tooth extraction; (**c**) Alveolus cleaning for natural clot formation. Experimental Group: (**d**) Alveolus cleaning; (**e**) CaSO_4_ + PRGF graft; (**f**) Graft in the alveolus.

**Figure 2 materials-14-01126-f002:**
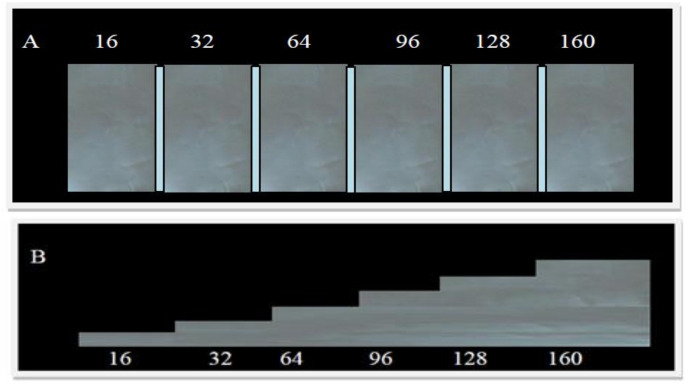
Grey Standardized Aluminum Scale (**A**) frontal view (**B**) lateral view.

**Figure 3 materials-14-01126-f003:**
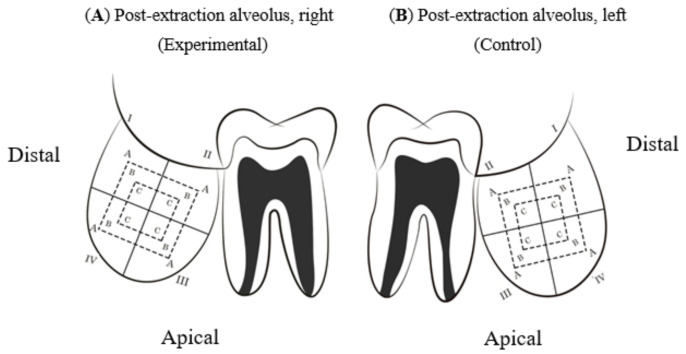
Bone regeneration diagrams for post-extraction alveoli. (**A**) EG, (**B**) CG.

**Figure 4 materials-14-01126-f004:**
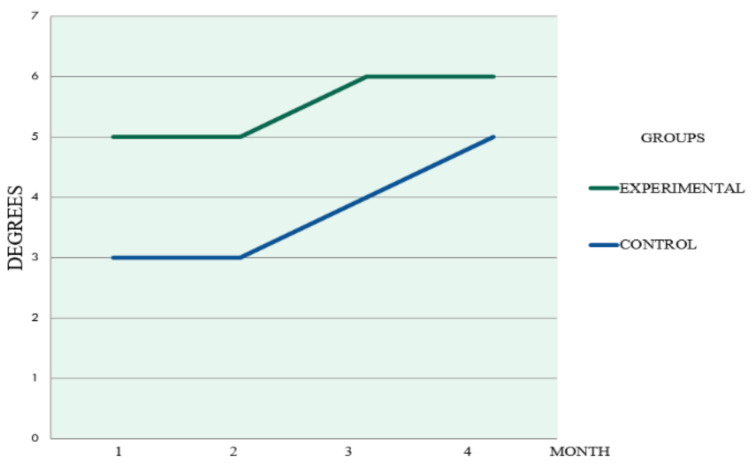
Monthly Ro/BR average post-extraction comparing EG and CG.

**Figure 5 materials-14-01126-f005:**
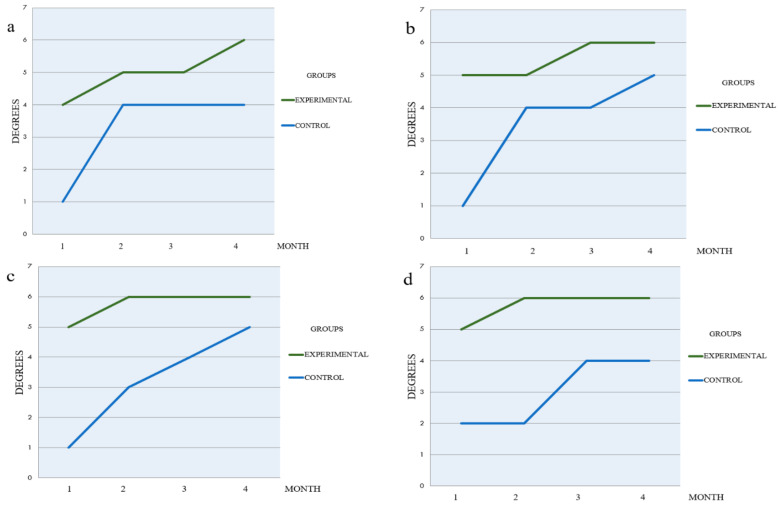
Quadrant Ro/BR 1-4, month evaluation (**a**) quadrant I (**b**) quadrant II (**c**) quadrant III (**d**) quadrant IV.

**Figure 6 materials-14-01126-f006:**
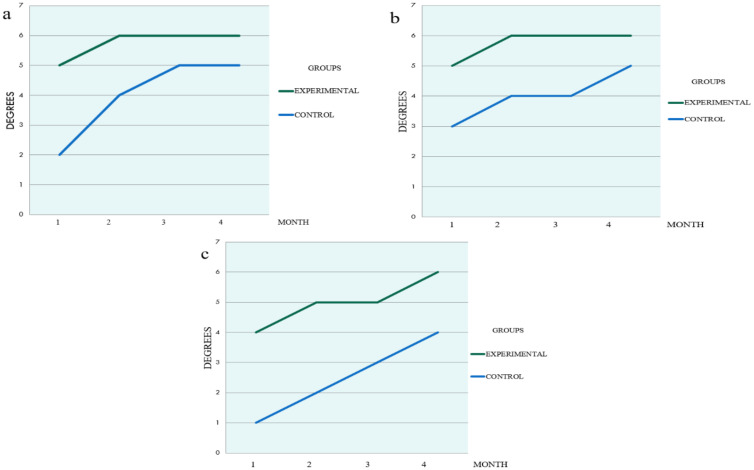
Areas Ro/BR 1-4, month evaluation (**a**) area A (**b**) area B (**c**) area C.

**Table 1 materials-14-01126-t001:** Bone RO percentages, where number 1 represents the lowest and 6 the highest.

Degree BR	Grey Scale (Steps on Tool)	Bone Regeneration (%)
1	16	1–16.66%
2	32	16.67–33.32%
3	64	33.33–49.98%
4	96	49.99–66.64%
5	128	66.65–83.3%
6	160	83.4–100%

**Table 2 materials-14-01126-t002:** Intra-group Friedman’s non-parametric test statistical result throughout 1–4 months post-extraction.

*p* Value	Month	Ro/BR
0.0001	1	1.35
EG	2	2.15
-	3	2.95
-	4	3.55
0.0001	1	1.20
CG	2	1.90
-	3	3.15
-	4	3.75

**Table 3 materials-14-01126-t003:** Non-parametric test of the sign comparing quadrants.

	Month 1 CG–EG	Month 2 CG–EG	Month 3 CG–EG	Month 4 CG–EG
*p* Value I	0.004	0.002	0.004	0.002
*p* Value II	0.004	0.002	0.002	0.002
*p* Value III	0.002	0.002	0.002	0.002
*p* Value IV	0.002	0.002	0.002	0.002

**Table 4 materials-14-01126-t004:** Non-parametric test of the sign comparing the areas.

	Month 1 CG–EG	Month 2 CG–EG	Month 3 CG–EG	Month 4 CG–EG
*p* Value A	0.002	0.002	0.004	0.016
*p* Value B	0.002	0.002	0.002	0.002
*p* Value C	0.002	0.002	0.002	0.002

## Data Availability

The protocol is registered in the ISRCTN data base with the number: 14620180, and can be found at https://doi.org/10.1186/ISRCTN14620180.

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
