# Peer review of "Calcium Sulfate and Plasma Rich in Growth Factors Enhance Bone Regeneration after Extraction of the Mandibular Third Molar: A Proof of Concept Study"

_materials, 2021, doi:10.3390/ma14051126_

Round 1

Reviewer 1 Report

Dear authors,

I would like to congratulate you for making this research in this COVID period. Your idea of using calcium sulphate and PRGF is innovative. Definitely, we need new materials to use it in bone regeneration process and enhance formation of new bone as soon as possible.

After reviewing your paper, I will have to reject it because you have a really low number of patients. If you are able to cover at least 40 patients with a follow-up of 1 year, I will be gladly review, once again your paper. In the end, I left you some suggestions for your future paper.

I wish you best regards and lots of health.

Reviewer

Suggestions

Title: The title should be changed as follows: "Calcium Surface and Plasma Rich in Growth Factors enhance bone regeneration after extraction of the Mandibular third molar: a proof of concept study"

Abstract: lots of typos. Please review the english and semantics. Also, you have only 10 patients included, which is quite low. You did a huge number of x-rays.

Introduction: Serious problems on English and semantics. Please, revise it.

Revise the introduction. You insert chapters, which is not accordingly to a scientific paper. You have 55 references which is a high number.

What is the aim of this paper?

M&M: What kind of study is this? Is it an RCT?

Still low number of patients. This is a big bias for you.

Figure 2. You have a figure and a table? Redone it.

Results: Figure 4. Low quality images. I cannot read it. Redone it.

Author Response

Response to Reviewer 1 Comments

Thank you very much for taking the time to review our work and especially for the suggestions made, which we will certainly attend to with the aim of improving it.

I would like to congratulate you for making this research in this COVID period. Your idea of using calcium sulphate and PRGF is innovative. Definitely, we need new materials to use it in bone regeneration process and enhance formation of new bone as soon as possible. After reviewing your paper, I will have to reject it because you have a really low number of patients. If you are able to cover at least 40 patients with a follow-up of 1 year, I will be gladly review, once again your paper. In the end, I left you some suggestions for your future paper. I wish you best regards and lots of health.

Reviewer

Suggestions

Title: The title should be changed as follows: "Calcium Surface and Plasma Rich in Growth Factors enhance bone regeneration after extraction of the Mandibular third molar: a proof of concept study"

Response: Thank you for the suggestion, we will ask the other reviewer if it´s possible.

Abstract: lots of typos. Please review the English and semantics. Also, you have only 10 patients included, which is quite low. You did a huge number of x-rays.

Response:

Typos were reviewed.

The sample number is justified because it was difficult to find and retain the number of patients who met the inclusion criteria and who remained throughout the assessment time. Number of X-rays were the only way to evaluated RO/BR. And demonstrate the reabsorption of the biomaterial.

Introduction: Serious problems on English and semantics. Please, revise it.

Response:

We hope that the changes will be satisfactory.

Revise the introduction. You insert chapters, which is not accordingly to a scientific paper. You have 55 references which is a high number.

Response:

The insertion of chapters was removed.

What is the aim of this paper?

Response:

The aim of this study was to evaluate the mixture of Calcium Sulfate and Plasma Rich in Growth Factors (CaSO4+PRGF) as a bone-graft-substitute in extracted mandibular third molars (MTM) alveoli, during a 4-month period.  

M&M: What kind of study is this? Is it an RCT?

Response: The study was an analytical, prospective, experimental and longitudinal. this data was added at the beginning of the material and methods.

Still low number of patients. This is a big bias for you.

Response:

The sample number is justified because it was difficult to find and retain the number of patients who met the inclusion criteria and who remained throughout the assessment time.

Figure 2. You have a figure and a table? Redone it.

Response:

Figure 2 was redone, and figure and table separated for a better comprehension.

Results: Figure 4. Low quality images. I cannot read it. Redone it.

Response:

Figure 4 was redone. Now figure 4 corresponds to the monthly comparison of the EG and CG, figure 5 represents quadrants monthly comparison (I, II, II, IV) and figure 6 is the areas monthly comparison (A, B, C).

Reviewer 2 Report

The authors presented an interesting topic related to bone regeneration with calcium sulfate and PRGF. The topic is interesting, however the manuscript requires several corrections and additions:

In the introduction the authors should also mention the alternative materials and tissue engineering-stem Cell application used in the regeneration of the alveolar bone, e.g Nanomaterials 2020, 10, 1216; doi:10.3390/nano10061216.

Which manufacturer's CaSO4 was used? There is no such information in Materials and Methods.

Whether all the procedures were performed by the same operator?

Did the authors apply an external cooling during a low speed piece use?

Did patients use antiseptics in the postoperative period?

The discussion needs to be expanded. Present results should be compared with other materials and techniques. The sentence (lines 241-242) is not true.

References need updating.

Author Response

Response to Reviewer 2 Comments

Thank you very much for taking the time to review our work and especially for the suggestions made, which we will certainly attend to with the aim of improving it.

The authors presented an interesting topic related to bone regeneration with calcium sulfate and PRGF. The topic is interesting, however the manuscript requires several corrections and additions:

In the introduction the authors should also mention the alternative materials and tissue engineering-stem Cell application used in the regeneration of the alveolar bone, e.g Nanomaterials 2020, 10, 1216; doi:10.3390/nano10061216.

Response:

In accordance with the suggestion, a review of different materials used to promote bone regeneration was carried out, which also served to nourish the discussion.

Which manufacturer's CaSO4 was used? There is no such information in Materials and Methods.

Response:

The CaSO4 used in the study was manufactured by MDC® Dental, Zapopan, Jalisco, Mexico.

Whether all the procedures were performed by the same operator?

Response:

All procedures were performed by the same operator (PRGF preparation, surgical procedure). This sentence was added to M & M section.

Did the authors apply an external cooling during a low speed piece use?

Response:

A low speed hand-piece (Medidenta® Las Vegas, Nevada, USA) with sterilized water irrigation and 703 milling bur (Hager & Mesinger® GMBH, Deu NEUSS, Germany).

Did patients use antiseptics in the postoperative period?

Response:

In the post-operative period, patients applied Bexident Gums gel (ISDIN®, Barcelona, Catalonia, Spain) after toothbrushing (3 times a day) for one month.

The discussion needs to be expanded. Present results should be compared with other materials and techniques. The sentence (lines 241-242) is not true.

Response:

The discussion was expanded.

The assertion made in these sentences was justified on the basis of the literature.

References need updating.

Response:

New references were added and served to broaden the introduction and discussion.

Reviewer 3 Report

The presented publication takes up a current hypothesis for the influence of Calcium Sulfate and PRGF on Bone Regeneration within Alveoli of Mandibular Third Molars. The reviewed study aims to assess the mixture of Calcium Sulfate and Plasma Rich in Growth Factors (CaSO4+PRGF) as a bone-graft-substitute in extracted mandibular third molars (MTM) alveoli, during a 4-month period. Using the means of a non-parametric test of the signs, the experimental group (EG) showed significant difference of Ro/BR between groups p=0.002 (p<0.05). According to quadrants and areas, significant differences were observed in all quadrants and areas p=0.002 (p<0.05) except in area A, in month 4 p=0.016 which could be explained due to it being the closest to native bone. In conclusion the authors came to the result experimental group (EG) CaSO4+PRGF showed a higher degree of bone regeneration.

The authors used a definition of PRGF as “an endogenous therapeutic technology used in regenerative medicine, thanks to its potential for stimulation and acceleration of tissue healing, through the growth factors found in platelets, which are universal initiators of almost every regeneration process”.

Indeed, in dentistry, the first studies with PRF or with PRGF were in the treatment of post-extraction defects, but the used definition is not in accordance with the obvious definition of PRF or PRGF. In recent years, platelet-rich fibrin (PRF) has gained wide attention for its utilization as a biocompatible regenerative material not only in the dental field but also in medical fields. Platelet concentrates are used to enhance osseous tissue healing in oral and craniofacial surgery. They can stimulate bone regeneration with minimum inflammatory response and unwanted complications. Leukocyte- and platelet-rich fibrin (L-PRF) is one of the four main families of platelet concentrates for surgical use. L-PRF is frequently used in oral and maxillofacial surgery as a surgical adjuvant to improve healing and promote tissue regeneration. The L-PRF technology is very simple and inexpensive (particularly in comparison to the many kinds of platelet-rich plasma PRP available on the market), and the method was developed as an open-access system.

The authors present a practice-oriented system for the generation of a fibrin matrix, which is obtained purely from the patient's own platelets. Obviously, predictable results with the body's own growth factors were achieved without additional additives, but the authors were not evaluating how changing some parameters of the L-PRF protocol may influence its biological signature, independently from the characteristics of the centrifuge considering:

  • the impact of the centrifuge characteristics
  • the impact of the vibrations on the fibrin polymerization and cell content
  • the impact of different protocol, different PRF, different growth factors content and release

Obviously, the authors are not aware of the current literature in the field, a quick pubmed search should be sufficiently.

As a conclusion, it is clearly proven that the different L-PRF protocols and different centrifuge characteristics and centrifugation protocols have a very significant impact on the cell, growth factors and fibrin architecture of a L-PRF clot and membrane and that any modification of the original L-PRF material and method shall be clearly investigated and identified. The authors should separately from the used in the publication methods differentiate their results in order to avoid confusion and inaccurate conclusions.

The usage of these concentrates was derived from the high content of growth factors that can be liberated from platelets at the time of tissue damage; these growth factors are essential for hard and soft tissue repair mechanisms. Among the advantages of platelet concentrates, their safety as an autologous source helps enhance early stability of grafts. Blood concentrates are prepared from the patient's own peripheral blood. This bioactive autologous system optimizes the success of dental implants by supporting the patient's own regeneration. The use of autologous blood concentrates is even more important in periodontology, e.g. to support the preservation of the tooth regeneratively.

Author Response

Response to Reviewer 3 Comments

Thank you very much for taking the time to review our work and especially for the suggestions made, which we will certainly attend to with the aim of improving it.

The presented publication takes up a current hypothesis for the influence of Calcium Sulfate and PRGF on Bone Regeneration within Alveoli of Mandibular Third Molars. The reviewed study aims to assess the mixture of Calcium Sulfate and Plasma Rich in Growth Factors (CaSO4+PRGF) as a bone-graft-substitute in extracted mandibular third molars (MTM) alveoli, during a 4-month period. Using the means of a non-parametric test of the signs, the experimental group (EG) showed significant difference of Ro/BR between groups p=0.002 (p<0.05). According to quadrants and areas, significant differences were observed in all quadrants and areas p=0.002 (p<0.05) except in area A, in month 4 p=0.016 which could be explained due to it being the closest to native bone. In conclusion the authors came to the result experimental group (EG) CaSO4+PRGF showed a higher degree of bone regeneration.

The authors used a definition of PRGF as “an endogenous therapeutic technology used in regenerative medicine, thanks to its potential for stimulation and acceleration of tissue healing, through the growth factors found in platelets, which are universal initiators of almost every regeneration process”.

Indeed, in dentistry, the first studies with PRF or with PRGF were in the treatment of post-extraction defects, but the used definition is not in accordance with the obvious definition of PRF or PRGF. In recent years, platelet-rich fibrin (PRF) has gained wide attention for its utilization as a biocompatible regenerative material not only in the dental field but also in medical fields. Platelet concentrates are used to enhance osseous tissue healing in oral and craniofacial surgery. They can stimulate bone regeneration with minimum inflammatory response and unwanted complications. Leukocyte- and platelet-rich fibrin (L-PRF) is one of the four main families of platelet concentrates for surgical use. L-PRF is frequently used in oral and maxillofacial surgery as a surgical adjuvant to improve healing and promote tissue regeneration. The L-PRF technology is very simple and inexpensive (particularly in comparison to the many kinds of platelet-rich plasma PRP available on the market), and the method was developed as an open-access system.

The authors present a practice-oriented system for the generation of a fibrin matrix, which is obtained purely from the patient's own platelets. Obviously, predictable results with the body's own growth factors were achieved without additional additives, but the authors were not evaluating how changing some parameters of the L-PRF protocol may influence its biological signature, independently from the characteristics of the centrifuge considering:

  • the impact of the centrifuge characteristics
  • the impact of the vibrations on the fibrin polymerization and cell content
  • the impact of different protocol, different PRF, different growth factors content and release

Obviously, the authors are not aware of the current literature in the field, a quick pubmed search should be sufficiently.

As a conclusion, it is clearly proven that the different L-PRF protocols and different centrifuge characteristics and centrifugation protocols have a very significant impact on the cell, growth factors and fibrin architecture of a L-PRF clot and membrane and that any modification of the original L-PRF material and method shall be clearly investigated and identified. The authors should separately from the used in the publication methods differentiate their results in order to avoid confusion and inaccurate conclusions.

The usage of these concentrates was derived from the high content of growth factors that can be liberated from platelets at the time of tissue damage; these growth factors are essential for hard and soft tissue repair mechanisms. Among the advantages of platelet concentrates, their safety as an autologous source helps enhance early stability of grafts. Blood concentrates are prepared from the patient's own peripheral blood. This bioactive autologous system optimizes the success of dental implants by supporting the patient's own regeneration. The use of autologous blood concentrates is even more important in periodontology, e.g. to support the preservation of the tooth regeneratively.

Response:

We agree that there are other protocols, but the study is based on the use of a standardized protocol of Anitua's PRFC as a scaffold for the CaSO4 spheres. However, we do not rule out the possibility of doing a new study including the L-PRF in order to be able to compare the two methods.

Reviewer 4 Report

The paper is on the use of patelet rich fibrin for the preservation of extractive sockets in the third molars.

The abstract should be without headings as requested in the guidelines of the journal.

Line 23: please change evaluated with evaluate.

Please, conclusion should be improved: “EG CaSO4+PRGF showed a higher degree of bone regeneration” I suggest to terminate the sentence with “if compared with…”

Introduction

Line 36: “since it begins from the momento it is performed”: the sentence should be improved with more scientific “sound”. Check for the word “memento” that it could be changed in “at the time”.

In dentistry, bone resorption after tooth extraction is of great importance, since it begins from the momento it is performed, normally characterized by a height and width of 25% in the first year and 40% in 3 years.

The sentence should be improved since “great importance” sounds too unscientific.

“height and width of 25% in the first year and 40% in 3 years.” Please carefully explain what 25% represents in terms of height and width.

The overall introduction section should be improved: I suggest to remove “bold” character of the subheading “Grafts can be classified into four groups:” to obtain a unique paragraph.

Line 51 - Extraorally, it can be taken from the iliac crest, femur, tibia, fibula, humerus, ribs and calotta. It´s advantages are that avoids skin scars and proximity to the recipient area. (Please invert the sequence of the sentences).

Line 76 Please change “between the 3rd-4th” with “in 3-4 weeks”.

The historical details from the last paragraph at lines 63-67 can be removed.

The aim of the paper should be expressed in one clear sentence which is missing in the article. Please add one at the proper time in the introduction section.

Materials and Methods

When citing a product, it should be followed by some mandatory details of the producer: e.g. ProductName (Producer, City, Country).

Please clearify that NOM-013-SSA2-2006 is a kind of protocol and of which reality.

Mucoperiostic can be changed in full-thickness flap.

Please provide SPSS details of the version, producer, etc.

Line 171 all the decimal numbers have to be separated by a “.” Instead of a “,”.

Results

Figure 3: is it conventional that the areas I, II, III and IV are not in clockwise order, but they are inverted: I,II, IV and III clockwisely?

Figure 4: please remove “rd” from the week’s number and fix the sentences. This should be done in the whole manuscript.

For an easier comprehension, figure 4 should be numbered from the very first image which is without it’s own letter. Please fix it.

Discussion

Discussion should be improved since it is very short if compared with the introduction. Maybe you can move here the last two paragraphs of the introducion.

Line 251 “sities” should be fixed in “sites”.

Line 260 “mucoperiosistic flap” should be fixed in “full-thickness flap”.

Leukocyte- and platelet-rich fibrin (L-PRF) is an autologous platelet concentrate rich in growth factors and plasma proteins, obtained by centrifugation of patient whole blood, and widely used in oral surgery, why the authors do not considerate this kind of autologous graft? It can be used the following article for further informations:

Serafini G, Lollobrigida M, Fortunato L, et al. Postextractive Alveolar Ridge Preservation Using L-PRF: Clinical and Histological Evaluation. Case Rep Dent. 2020 Jun 11;2020:5073519. doi: 10.1155/2020/5073519.

Other authors recently published a paper on the alveolar socket preservation of the third molars made by the grafting of dentinal grinded fragments: it could be interesting to evaluate the differences between the two methods even in terms of costs and difficult in obtain the materials. Do the author suggest the use of both procedures in association?

The following paper analyzes the histologic and histomorphometric aspects of bone regeneration using bovine grafting material, which are the differences between: Guarnieri R, Belleggia F, DeVillier P, Testarelli L. Histologic and Histomorphometric Analysis of Bone Regeneration with Bovine Grafting Material after 24 Months of Healing. A Case Report. J Funct Biomater. 2018 Aug 8;9(3):48. doi: 10.3390/jfb9030048.

Can you underline which are the substantial differences in terms of histology and histomorphometrics of the grafted site?

References should be present on the most common international indexing databases like Pubmed, Scopus and WoS. International literature is usually in american english and all the references present in the paper are from latin databases. Please fix it.

Author Response

Response to Reviewer 4 Comments

Thank you very much for taking the time to review our work and especially for the suggestions made, which we will certainly attend to with the aim of improving it.

The paper is on the use of patelet rich fibrin for the preservation of extractive sockets in the third molars.

The abstract should be without headings as requested in the guidelines of the journal.

Response:

Headings were removed from the abstract.

Line 23: please change evaluated with evaluate.

Response: Change made.

Line 22: Signs Test was used to evaluate Radiopacity/Bone.

Please, conclusion should be improved: “EG CaSO4+PRGF showed a higher degree of bone regeneration” I suggest to terminate the sentence with “if compared with…”

Response:

EG CaSO4+PRGF showed a higher degree of bone regeneration if compared to CG.

Introduction.

Line 36: “since it begins from the momento it is performed”: the sentence should be improved with more scientific “sound”. Check for the word “memento” that it could be changed in “at the time”.

Response:

The meaning of the paragraph was changed, and the word "moment" was removed.

Line 34-36: In maxilla’s, 25% of bone resorption after tooth extraction, characterized by a height and width lose is show in the first year and 40% in 3, beginning mainly in the buccal surface, as it is thinner and more fragile.

In dentistry, bone resorption after tooth extraction is of great importance, since it begins from the momento it is performed, normally characterized by a height and width of 25% in the first year and 40% in 3 years.

“height and width of 25% in the first year and 40% in 3 years.” Please carefully explain what 25% represents in terms of height and width.

The sentence should be improved since “great importance” sounds too unscientific.

Response: The paragraph was changed

Line 34-36: In maxilla’s, 25% of bone resorption after tooth extraction, characterized by a height and width lose is show in the first year and 40% in 3, beginning mainly in the buccal surface, as it is thinner and more fragile.

The overall introduction section should be improved: I suggest to remove “bold” character of the subheading “Grafts can be classified into four groups:” to obtain a unique paragraph.

Response:

line 46: Grafts can be classified as: Autografts: (same individual)…

Line 51 - Extraorally, it can be taken from the iliac crest, femur, tibia, fibula, humerus, ribs and calotta. It´s advantages are that avoids skin scars and proximity to the recipient area. (Please invert the sequence of the sentences).

Response: The sequence of the phrases was inverted.

Line 47-51: Intraorally, they are obtained from chin, retromolar region, nasal spine, exostoses, ascending mandibular ramus and the lower edge of the jaw. It´s advantages are that avoids skin scars and proximity to the recipient area. Extraorally, it can be taken from the iliac crest, femur, tibia, fibula, humerus, ribs and calotta. However, this type of graft increases morbidity and requires longer surgical time.

Line 76 Please change “between the 3rd-4th” with “in 3-4 weeks”.

Response: Change made …. line 68: 3-4 weeks.

The historical details from the last paragraph at lines 63-67 can be removed.

Response: The paragraph (lines 63-67) was removed.

The aim of the paper should be expressed in one clear sentence which is missing in the article. Please add one at the proper time in the introduction section.

Response: added in the abstract and introduction sections.

The aim of this study was to evaluate the mixture of Calcium Sulfate and Plasma Rich in Growth Factors (CaSO4+PRGF) as a bone-graft-substitute in extracted mandibular third molars (MTM) alveoli, during a 4-month period.

Materials and Methods

When citing a product, it should be followed by some mandatory details of the producer: e.g. ProductName (Producer, City, Country).

Response: The data for the products cited in the study were completed.

Line108-109: x-ray orthopantomography (Orthopantomograph OP200 D® Orthoceph®, Nahkelantie, Tuusula, Finland).

Line 114: CaSO4 (MDC® Dental, Zapopan, Jalisco, México).

Line 115: dry heat sterilization (LORMA M-08® Mexico City, Mexico).

Line 116-117: BTI© Plasma Transfer Device kit (PTD®, BTI Biotechnology Institute, Vitoria-Gasteiz, Álava, Spain).

Line 117-118: Endoret® BTI (BTI Biotechnology Institute, Vitoria-Gasteiz, Álava, Spain) centrifuge

Line 124-125: anesthesia with Dentocain (mepivacaine & epinephrine 2%, Zeyco®, Zapopan Jalisco, México).

Line 127: low speed piece (Medidenta® Las Vegas, Nevada, USA).

Line 128: 703 milling burs (Hager & Mesinger® GMBH, Deu NEUSS, Germany).

Line 129: elevation (No.3 elevator®, Dental USA, Chicago, Illinois, USA).

Line 130: (Lucas curette®, Dental USA, Chicago, Illinois, USA).

Line 132: 4-0 silk (Ethicon® US LLC, Cincinnati, OH, USA).

Line 155-156: (Satelec X-Mind DC® X-ray equipment, Burnlea Grove, Birmingham, England).

Line 207: SPSS package version 20 statistical (IBM® New York, NY, USA).

Please clearify that NOM-013-SSA2-2006 is a kind of protocol and of which reality.

Response:

Line 107-108: NOM-013-SSA2-2006 (Norma Oficial Mexicana for prevention and control of oral diseases).

Mucoperiostic can be changed in full-thickness flap.

Response: Change made

Line 126: A full-thickness triangular flap was created.

Line 152: (a) Full-Thickness triangular flap.

Please provide SPSS details of the version, producer, etc.

Response:

Line 207: SPSS package version 20 statistical (IBM® New York, NY, USA).

Line 171 all the decimal numbers have to be separated by a “.” Instead of a “,”.

Response: Change made.

Line 208: p<0.05.

Results

Figure 3: is it conventional that the areas I, II, III and IV are not in clockwise order, but they are inverted: I,II, IV and III clockwisely?

Response: Change made

Line 182: dividing them into four quadrants I to IV. I and IV represent distal quadrants and II and III mesial. Look at figure 3.

Figure 4: please remove “rd” from the week’s number and fix the sentences. This should be done in the whole manuscript.

Response:

 The "rd" of the months were eliminated and the phrases were changed throughout the manuscript.

Figure 4. Monthly Ro/BR average post-extraction comparing EG and CG.

For an easier comprehension, figure 4 should be numbered from the very first image which is without it’s own letter. Please fix it.

Response:

For an easier comprehension, figure 4 the change was made to separate the images according to the monthly bone regeneration, as well as to group the quadrants (I, II, II, IV) (figure 5) and the areas (A, B, C) (figure 6).

Discussion

Discussion should be improved since it is very short if compared with the introduction. Maybe you can move here the last two paragraphs of the introducion.

Response:

Discussion was extended and improved

Line 251 “sities” should be fixed in “sites”.

Response:

Change made…. Line 344: extraction sites in a

Line 260 “mucoperiosistic flap” should be fixed in “full-thickness flap”.

Response:

Change made…. Line 353: full-thikness flaps and autologous

Leukocyte- and platelet-rich fibrin (L-PRF) is an autologous platelet concentrate rich in growth factors and plasma proteins, obtained by centrifugation of patient whole blood, and widely used in oral surgery, why the authors do not considerate this kind of autologous graft? It can be used the following article for further informations:

Serafini G, Lollobrigida M, Fortunato L, et al. Postextractive Alveolar Ridge Preservation Using L-PRF: Clinical and Histological Evaluation. Case Rep Dent. 2020 Jun 11;2020:5073519. doi: 10.1155/2020/5073519.

Response:

Serafini et al. (2020) reported a case of post-extraction alveolar ridge preservation with placement of three previously bent L-PRF membranes and clinical controls at 14, 28 and 60 days. Three months later, histomorphometric analysis of the grafted site was performed and newly formed trabecular bone was observed. They suggest that the use of L-PRF could achieve good results in terms of dimension, bone quality and soft tissue healing.

Other authors recently published a paper on the alveolar socket preservation of the third molars made by the grafting of dentinal grinded fragments: it could be interesting to evaluate the differences between the two methods even in terms of costs and difficult in obtain the materials. Do the author suggest the use of both procedures in association?

We are of course, open to further studies comparing CaSo4/PRGF with L-PRF alone or in combination for bone regeneration. However, at the moment we are limited to the facilities of the surgery clinic of the Faculty of Dentistry where we generally have enough patients to develop research protocols.

The following paper analyzes the histologic and histomorphometric aspects of bone regeneration using bovine grafting material, which are the differences between: Guarnieri R, Belleggia F, DeVillier P, Testarelli L. Histologic and Histomorphometric Analysis of Bone Regeneration with Bovine Grafting Material after 24 Months of Healing. A Case Report. J Funct Biomater. 2018 Aug 8;9(3):48. doi: 10.3390/jfb9030048.

Can you underline which are the substantial differences in terms of histology and histomorphometrics of the grafted site?

Response:

There is no possibility of obtaining any histological sample from the patients, firstly because we cannot access the data stored at the University and secondly because it could generate bioethical conflicts. We recognize that it would be ideal.

References should be present on the most common international indexing databases like Pubmed, Scopus and WoS. International literature is usually in american english and all the references present in the paper are from latin databases. Please fix it.

Response:

references have been fixed and updated.

Round 2

Reviewer 1 Report

Dear authors,

I congratulate you for improvements that you brought to your manuscripts. There are still flaws in the paper; here are as follows:

1) Title:  should be changed as follows: "Calcium Surface and Plasma Rich in Growth Factors enhance bone regeneration after extraction of the Mandibular third molar: a proof of concept study" 

2) Introduction: You mentioned 58 references. This is not possible.

3) M&M: you still have a low number of patients. Which shows to me that it is still not relevant the benefits of your product. Also, there is a lack of registration for this experiment in the RCT database.

4) Too many tables and figures which makes your paper difficult to understand.

5) Major language revision.

Author Response

Reviewer 1

Good morning and happy new year

We appreciate very much that you take time at this time to review our article.

1. Regarding the title, you suggest:

Calcium surface and plasma rich in growth factors improve bone regeneration after extraction of the third mandibular molar: a proof of concept study.

I ask you if it could be:

Calcium sulphate and plasma rich in growth factors improve bone regeneration after extraction of the third mandibular molar: a proof of concept study

2.  Introduction: You mentioned 58 references. This is not possible.

References in the introduction were reduced to 37

3. M&M: you still have a low number of patients. Which shows to me that it is still not relevant the benefits of your product. Also, there is a lack of registration for this experiment in the RCT database.

The study was definitely for convenience because the inclusion criteria ruled out most of the 46 patients reviewed.

On the other hand, the data in line 111 was added.

4. Too many tables and figures which makes your paper difficult to understand.

We believe that none of the tables or figures can be removed as the information may be incomplete.
Furthermore, dividing them up makes it easier to understand the results.

5. Major language revision.

The article will be sent immediately for English review to the "Author Services" of the journal.

Reviewer 2 Report

The reviewer accepts the changes made by the authors. 

Author Response

Dear Reviewer
We very much appreciate the time you took to make the revisions as well as the suggestions you gave us to improve the writing. Merry Christmas.

Reviewer 4 Report

The authors made an extensive enrichment of the paper as requested.

No further changes are required.

Author Response

(The authors gave the same response as above.)
